# Extra-Skeletal Effects of Vitamin D

**DOI:** 10.3390/nu11071460

**Published:** 2019-06-27

**Authors:** Rose Marino, Madhusmita Misra

**Affiliations:** Pediatric Endocrine Unit, Massachusetts General Hospital and Harvard Medical School, Boston, MA 02114, USA

**Keywords:** vitamin D, type 1 diabetes, type 2 diabetes, metabolic syndrome, autoimmune, children, cancer

## Abstract

The vitamin D receptor is expressed in multiple cells of the body (other than osteoblasts), including beta cells and cells involved in immune modulation (such as mononuclear cells, and activated T and B lymphocytes), and most organs in the body including the brain, heart, skin, gonads, prostate, breast, and gut. Consequently, the extra-skeletal impact of vitamin D deficiency has been an active area of research. While epidemiological and case-control studies have often suggested a link between vitamin D deficiency and conditions such as type 1 and type 2 diabetes, connective tissue disorders, inflammatory bowel disorders, chronic hepatitis, food allergies, asthma and respiratory infections, and cancer, interventional studies for the most part have failed to confirm a causative link. This review examines available evidence to date for the extra-skeletal effects of vitamin D deficiency, with a focus on randomized controlled trials and meta-analyses.

## 1. Introduction

While the skeletal effects of vitamin D are well recognized and described extensively in the literature [1,2,3], its extra-skeletal effects have been subject to some controversy with conflicting data reported, particularly for case-control or epidemiologic vs. prospective and interventional studies. This review aims to summarize and synthesize data regarding many extra-skeletal effects of vitamin D. Given the extensive literature reported in this area over the last two decades, we have discussed key papers that illustrate variations in data reported from the various kinds of studies, with a focus on randomized controlled trials (RCTs) and meta-analyses of existing studies.

Vitamin D_3_ (cholecalciferol) is synthesized primarily in the skin on exposure to ultraviolet radiation, while vitamin D_2_ (ergocalciferol) is derived from plant sources. 7-dehydrocholesterol (provitamin D; present in the stratum basale and stratum spinosum of the epidermis), is converted to previtamin D on exposure to ultraviolet radiation-B (UV-B), which is then isomerized to vitamin D. Vitamin D passes into dermal capillaries and is carried by vitamin D binding protein (DBP) to the liver, where microsomal vitamin D 25-hydroxylase catalyzes its conversion to 25-hydroxy vitamin D [25(OH)D], the storage form of vitamin D. 25(OH)D is what is reported when we ask for levels of vitamin D, and most assays report both 25(OH)D_2_ and 25(OH)D_3_. While controversy persists around the normative range for 25(OH)D levels, the Institute of Medicine has indicated that a 25(OH)D level at or above 20 ng/mL (50 nmol/L) is likely sufficient to optimize its skeletal effects [4]. Normative ranges for its possible extra-skeletal effects remain to be determined. Vitamin D supplements may contain either vitamin D_2_ or D_3,_ or alfacalcidiol (1-hydroxycholecalciferol). Overall, data suggest that vitamin D_3_ may be more effective in raising 25(OH)D levels than vitamin D_2_ (reviewed in [5]).

25(OH)D is transported to the kidney by DBP, where cytochromal 25-hydroxyvitamin D 1-α hydroxylase catalyzes its conversion to 1,25 dihydroxy vitamin D [1,25(OH)_2_D], the active form of vitamin D. The vitamin D receptor (VDR) is expressed in multiple cells including the osteoblasts, mononuclear cells, activated T and B lymphocytes and beta cells, and most organs in the body including the brain, heart, skin, gonads, prostate, breast, and gut. How vitamin D deficiency affects disease states in these multiple organ systems has been an active area of research. Repletion of vitamin D stores to mitigate and improve disease processes has been attempted in certain conditions, although there is a paucity of data to direct clear treatment protocols, especially in the pediatric population. 

## 2. Immune and Anti-Inflammatory Effects

There is evidence that vitamin D modulates B and T lymphocyte function [6,7,8,9], and vitamin D deficiency has been associated with conditions such as multiple sclerosis, type 1 diabetes (T1D), rheumatoid arthritis, systemic lupus erythematosus, dermatomyositis, inflammatory bowel disease, hepatitis, asthma and respiratory infections.

### 2.1. Type 1 Diabetes

Data regarding associations of 25(OH)D levels or vitamin D sufficiency/insufficiency/deficiency status are conflicting with some [10,11], but not all [12,13,14], studies reporting an association between low vitamin D status and occurrence of T1D. Pancreatic beta cells have VDRs and express 1-α hydroxylase (encoded by *CYP27B1*) [15], and the human insulin gene promoter has a vitamin D response element [16]. Further, vitamin D plays a role in T-cell regulatory responses and may protect beta cells from immune attack [17]. In addition, T1D patients are reported to have lower 25(OH)D levels compared to age matched controls [11,18,19]. Cooper et al. [20] linked the genetic determinants of circulating 25(OH)D (*DHCR7* and *CYP2R1*, which encode 7-dehydrocholesterol reductase and 25-hydroxylase) and vitamin D signaling in T cells (*CYP27B1*) with risk of T1D, while others (including a meta-analysis of nine studies with 1053 patients and 1017 controls) have linked specific polymorphisms of the VDR gene with risk for T1D [21]. Another study linked T-cell proliferation with DBP, and reported higher levels and frequencies of serum anti-DBP antibodies in patients with T1D vs. controls. This study postulated that DBP (expressed in α-cells of pancreatic islets) may be an autoantigen in T1D [22]. Further, lower maternal third trimester DBP levels and cord blood DBP levels have been associated with risk of T1D in offspring [23,24]. Further investigation into the role of CYP27B1 in immune cells such as monocytes, macrophages and T-cells is needed to better understand the role of vitamin D in the pathogenesis and perhaps prevention of T1D. Under this umbrella of research, prospective studies of vitamin D supplementation have attempted to elucidate causality of T1D and use of vitamin D as a potential therapy. 

Interventional studies: Table 1 describes details of some representative studies of vitamin D supplementation in T1D. A beneficial effect of cholecalciferol supplementation on regulatory T-cells (T-regs) has been reported, with an increase in T-reg percentage [25] and suppressive capacity [26], and reduced progression to undetectable C-peptide. Consistent with these findings, a meta-analysis of five observational studies reported a protective effect of vitamin D supplementation in early childhood against development of T1D with a dose response effect [27], while another reported that such supplementation may have prevented 27% of the predicted T1D cases in England and Wales in 2012 [28]. Similarly, a beneficial impact has been reported with respect to fasting C-peptide with lower daily insulin doses [29], peripheral vascular resistance and inflammatory renal markers [30]. However, many other studies have not demonstrated a beneficial effect of vitamin D supplementation in preventing or improving the course of T1D. Importantly, the prospective Environmental Determinants of Diabetes in the Young (TEDDY) Study demonstrated no benefit of maternal vitamin D supplementation during pregnancy on the risk of islet autoimmunity in the offspring [31]. Other studies have failed to demonstrate a beneficial impact on beta cell function [32], HbA1C levels [25,33,34] or insulin requirement [25,34]. While the reason for these conflicting results is unclear, one may speculate that differences in study design, sample size and vitamin D dosing may contribute. Overall, randomized controlled studies investigating vitamin D replacement in preventing T1D or treating at diagnosis to prolong endogenous insulin secretion are few. Vitamin D replacement is necessary in those deficient for optimal bone health. However, current data do not provide definitive evidence that supplementation will improve the inflammatory state in a clinically significant manner.

### 2.2. Multiple Sclerosis

The risk for multiple sclerosis (MS) is higher in people living above 35° latitudes than those who live below this latitude [53] (although effects were attenuated over time in this study), and studies have reported an association between risk for MS and lower sunlight exposure [54,55,56,57,58,59], vitamin D intake [60] and serum 25(OH)D levels [61,62,63,64]. Increased sun exposure in children has been linked to a lower risk of MS in studies out of Tasmania [57], twin studies where only one monozygotic twin developed MS [55], a study from Cuba, Martinique and Sicily [54], as well as studies in adults from Iran [58] and Sweden [59]. Other studies have linked the reduced risk of MS following sun exposure to the melanocortin 1 receptor genotype [56]. Further, polymorphisms in genes that impact 25(OH)D levels have been linked to risk of relapse in children with MS [65]. 

Munger et al. reported an inverse relationship between serum 25(OH)D and the risk of MS [61], and that before onset, MS risk decreases by 40% for every 20 ng/mL increase in 25(OH)D levels. In another study, women (but not men) with MS had lower 25(OH)D levels than controls 11 years after disease onset [62], with a 19% reduction in the odds for MS for every 4 ng/mL increase in winter 25(OH)D concentrations. Similar lower 25(OH)D in patients with MS versus controls has been reported in other studies for winter [63] or summer levels [64]. 

Data suggest that vitamin D sufficient states in the mother and infant may protect against MS [66]. Increased first trimester exposure to UVR reduced MS risk in the child in a dose dependent fashion [67]. Higher MS rates in children born in the summer or spring, and lower rates in those born in the fall may be related to maternal vitamin D insufficiency [68,69]. In the Nurses Health Study, women born to mothers with high vitamin D intake during pregnancy had a reduced risk of MS [70]. 

Interventional Studies: protective effects of vitamin D supplementation have been demonstrated against MS in a few studies [60]. The Nurses Health Study reported that women taking at least 400 IU of vitamin D supplements daily were at lower risk compared to those not on supplements [60,71]. A meta-analysis of 12 RCTs that included 950 adult patients with MS reported non-significant trends in improvement in annualized relapse rate Expanded Disability Status Scale scores and MRI findings in those who received vitamin D supplementation, although higher vitamin D doses (2850–10,400 IUs) performed significantly worse compared to lower doses (800–1000 IUs) [72]. One RCT did report an improvement in MRI lesions when vitamin D_3_ was employed as add-on treatment to interferon beta-1b in patients with MS [73]. Data are lacking for effects of vitamin D supplementation in children with MS. 

Overall, although case-control and epidemiological studies suggest that people with lower 25(OH)D levels may be at higher risk for MS, confounding factors, such as the association of higher latitudes (also a risk factor for MS) with lower 25(OH)D levels need to be considered. Interventional studies examining the preventive effect of vitamin D supplementation on the risk for developing MS are lacking. Existing RCTs in adult patients with established MS are not convincing for a significant role for vitamin D supplementation in reducing MS outcomes, although trends for a positive effect have been reported, particularly with low dose supplementation. Data to date indicate deleterious effects of high dose vitamin D supplementation on MS outcomes. RCTs assessing the impact of vitamin D supplementation on MS outcomes in children are lacking.

### 2.3. Rheumatoid Arthritis

One meta-analysis of 15 studies (including 1143 rheumatoid arthritis (RA) patients and 963 controls) confirmed lower 25(OH)D levels and a higher prevalence of vitamin D deficiency in patients compared with controls (55% vs. 33%) [74]. Similarly, a meta-analysis of three cohort studies that included 215,757 participants and 874 incident cases of RA reported that individuals in the highest quartile for vitamin D intake had a 24% lower risk of developing RA compared to those in the lowest quartile [75]. In addition, inverse associations of 25(OH)D levels with disease activity scores in RA patients have been reported [74,76,77,78,79]. Further, specific VDR polymorphisms have been demonstrated to contribute significantly to RA risk [80,81]. 

Interventional Studies: consistent with the findings from epidemiologic studies, protective effects of vitamin D supplementation have been demonstrated against RA in some studies [82]. Chandrashekara and Patted reported a reduction in disease severity scores following vitamin D replacement (60,000 IU of vitamin D weekly for six weeks followed by 60,000 IU monthly) over three months in RA patients who were vitamin D deficient (25(OH)D levels <20 ng/mL) and had persistent disease activity [83]. However, a meta-analysis that included five RCTs in patients with RA reported borderline significance for reduction in disease recurrence, but not for disease activity [84]. Calcitriol enhances inhibition of T cell activation by abatacept (CTLA-4-Ig), a CD28-ligand blocker to which patients with RA respond with variable efficacy [85], suggesting that vitamin D or calcitriol may be used as an adjunct therapy to improve efficacy of abatacept in RA patients. Overall, these data suggest that lower 25(OH)D levels may predict susceptibility to RA and its severity, but it is not clear that vitamin D supplementation reduces the risk of developing RA or its disease activity/severity. Further, data in children with RA are currently lacking.

### 2.4. Systemic Lupus Erythematosus (SLE) and Juvenile Dermatomyositis 

Many studies have reported inverse associations of 25(OH)D levels with the occurrence and severity of SLE and dermatomyosits in adults and children [86,87,88,89,90,91,92]. In children, greater disease severity of SLE is reported in those with 25(OH)D levels <20 ng/mL [88]. 

Interventional Studies: in contrast to other autoimmune conditions, several interventional studies have reported an improvement in markers of severity of disease following vitamin D supplementation in patients with SLE. One study in adolescents and adults with juvenile onset SLE reported an improvement in disease severity scores and fatigue in those randomized to 50,000 IU per week of vitamin D supplementation compared to placebo over 6 months [93]. Similar findings are reported in studies of adults with SLE. A reduction in inflammatory and hemostatic markers, and disease activity was reported in those randomized to 2000 IU of vitamin D daily compared to placebo for 12 months [94]. Further, a meta-analysis of three RCTs of vitamin D supplementation in patients with SLE reported a significant reduction in anti-dsDNA positivity with supplementation [84]. Another study reported an increase of T-regs and a decrease of effector Th1 and Th17 cells, memory B cells and anti-DNA antibodies in adults with SLE given 100,000 IU of vitamin D weekly for 4 weeks followed by 100,000 IU monthly for 6 months [95]. No flares were observed over the study duration [95]. 

Overall, these data suggest that low 25(OH)D levels are concerning for greater risk of developing SLE and increased severity of the condition, and a possible beneficial effect of vitamin D supplementation on disease severity, particularly when levels are low. However, more RCTs, including in children, are necessary to confirm these data. Of note, symptoms of SLE can worsen after UV radiation exposure which will otherwise increase the vitamin D status of the individual [96]. Thus, excessive sun exposure to improve 25(OH)D levels in patients with SLE may worsen symptoms of the disease.

### 2.5. Psoriasis

Keratinocytes express the vitamin D receptor (VDR), and vitamin D inhibits the growth of keratinocytes, stimulates them to differentiate, and helps maintain the cutaneous barrier integrity [97]. This has led to the use of topical vitamin D analogs to treat psoriasis [98]. However, no VDR genotype (other than the Taql polymorphsm in Caucasians) has been linked to an increased risk for psoriasis [99,100]. In addition, there is no correlation between the change in 25(OH)D post phototherapy and change in severity of symptoms of psoriasis [101,102]. 

Interventional Studies: a 6-month RCT of 60,000 IUs of vitamin D vs. placebo given every 2 weeks to 45 patients with psoriasis showed an increase in 25(OH)D levels in the active group associated with a reduction in the Psoriasis Area and Severity Index (PASI) [103]. Topical vitamin D analogs (calcipotriene/calcipotriol/maxacalcitol) given with topical betamethasone have also been shown to be effective in improving plaque psoriasis [104,105,106] by disrupting the IL-36 and IL-23/IL-17 positive feedback loop, key factors in the pathogenesis of psoriasis. 

Overall, while studies suggest an association between 25(OH)D levels and psoriasis, causation is yet to be established. Additionally, RCTs to date are promising for an impact of vitamin D on psoriasis severity, but studies are small, and more and larger RCTs are necessary to confirm these preliminary findings. Data in children are lacking.

### 2.6. Inflammatory Bowel Disease (IBD)

Protective effects of vitamin D supplementation have been demonstrated against IBD [107]. Vitamin D deficient and vitamin D receptor (*Vdr*) null mice are at high risk of IBD [108], with more severe disease and more spontaneous recurrences. Further, polymorphisms in the *Vdr* gene (TaqI) may confer susceptibility to IBD [109]. A meta-analysis of VDR genotyping in relation to IBD reported that ApaI polymorphism may increase the risk of Crohn’s disease, whereas the TaqI polymorphism may decrease the risk of ulcerative colitis, especially in Caucasians [110]. A connection has been proposed between seasonal vitamin D status and risk for both Crohn’s disease and ulcerative colitis [111,112], with an association between vitamin D deficiency and more complicated disease course. A recent meta-analysis reported lower 25(OH)D levels in patients with both Crohn’s disease and ulcerative colitis than controls, a two-fold risk of vitamin D deficiency in those with IBD, and greater severity of Crohn’s disease in those with vitamin D deficiency [113,114]. However, these studies are associative and confounded by the fact that malabsorption (particularly in Crohn’s disease) may contribute to lower 25(OH)D levels, with more severe disease being associated with more malabsorption. 

Interventional studies: there are few randomized controlled trials that have examined the impact of vitamin D supplementation on disease occurrence or severity. One randomized controlled trial of 400 vs. 2000 IU of vitamin D supplementation in children and adolescents with IBD did report that the higher dose of vitamin D led to lower levels of pro-inflammatory markers [115]. Vitamin D has also been shown to exert marked anti-inflammatory effects on peripheral and intestinal CD4+ and CD8+ T cells of patients with inflammatory bowel disorders in vitro, and inhibit production of TH1 and TH17 cytokines in patients with Crohn’s disease in vivo [116]. 

Overall, while data to date suggest that vitamin D deficiency may impact disease course in patients with inflammatory bowel disorders, we are in need of robust RCTs that will help determine whether vitamin D supplementation is indeed efficacious in reducing disease severity in these patients, including in children.

### 2.7. Food Allergies

Although associations of low 25(OH)D levels with a risk of atopy, asthma and food allergies have been reported [117], recent birth cohort studies demonstrate no association of 25(OH)D levels (including antenatal vitamin D exposure) with the incidence of food allergy [118,119]. One meta-analysis of more than 5000 children did not find a significant association between vitamin D status and food allergy [120]. 

Interventional Studies: data for the role of vitamin D supplementation in preventing food allergies is inconsistent, and large trials are necessary to determine the role if any of vitamin D in reducing the risk for food allergies [121]. One RCT of 400 vs. 1200 IUs of vitamin D supplementation in 975 infants from the age of 2 weeks found no impact of higher-dose vitamin D supplementation on allergic sensitization or allergic diseases (food or aeroallergen) at 12-months of age [122]. Overall, current data do not support a role of vitamin D status in impacting the risk for food allergies in children, and non-interventional studies may be confounded by variables such as associated malabsorption.

### 2.8. Chronic Hepatitis B and C

Certain vitamin D receptor gene (*VDR* a/a) polymorphisms have been linked to greater severity of hepatitis B infection and a higher viral load [123]. In addition, polymorphisms in the T/T allele of exon 9 of the *VDR* gene (but not intron 8 polymorphisms) are associated with occult hepatitis B infection [124]. Further, a meta-analysis of seven studies involving 814 patients with chronic hepatitis B and 696 controls showed lower 25(OH)D levels in patients with chronic hepatitis B associated with higher viral loads [125]. Petta et al. [126] reported that lower 25(OH)D levels in chronic hepatitis C infection were related to decreased CYP27A1 expression, female gender, necrosis and inflammation, severe fibrosis and poor viral response to interferon based therapy. Of note, polymorphisms in the VDR gene have been associated with development of hepatocellular carcinoma in patients with hepatitis C [127]. 

Interventional Studies: in in vitro studies, 25(OH)D has been shown to inhibit hepatitis C virus production by suppressing apolipoprotein expression [128]. In 42 patients with recurrent hepatitis C infections treated with INF-α and ribavirin for 48 weeks, vitamin D deficiency was associated with an unfavorable response to antiviral medication, and 15 patients given oral vitamin D supplements had a better and more sustained response to antiviral treatment [129]. One short 6-week RCT of vitamin D supplementation vs. placebo in 54 patients with chronic hepatitis C with vitamin D deficiency demonstrated improved serum markers of hepatic fibrogenesis upon correction of 25(OH)D levels [130]. 

Overall, data suggest deleterious effects of vitamin D deficiency on the course of chronic hepatitis B and C infections, with some studies suggesting a possible role (and mechanism) for vitamin D supplementation in improving outcomes in these patients. However, we are in need of larger RCTs and of longer duration to effectively confirm these findings, and data in children are lacking.

### 2.9. Asthma and Respiratory Infections

Several studies have reported an association between vitamin D deficiency (25(OH)D < 20 ng/mL) and increased airway inflammation, decreased lung function, increased exacerbations, and poor prognosis in patients with asthma (reviewed in [131]).

Interventional Studies: a literature review of RCTs of vitamin D supplementation in children older than 2 years found a possible effect of vitamin D supplementation in improving bronchial asthma exacerbation [132,133], but no effect on the severity of asthma [132]. A subsequent meta-analysis that included 435 children (seven trials) and 658 adults (two trials) similarly reported a reduction in asthma exacerbations requiring systemic corticosteroids and the risk for an exacerbation requiring an emergency department visit or hospitalization in those who received vitamin D supplementation, but not on measures of severity (% predicted forced expiratory volume in one second or Asthma Control test Scores [134]. The improvement in asthma exacerbation has been attributed to a decrease in respiratory infections [12]. 

RCTs of vitamin D supplementation (300–1200 IUs per day) vs. placebo report a potential effect of vitamin D in reducing the risk for influenza infections during the winter months [135], although a subsequent analysis reported that benefits observed in the first month of supplementation were lost in the second month, and overall there was no difference among groups for influenza prevalence over the entire season [136]. A meta-analysis of 25 RCTs using vitamin D supplementation of any duration in individuals 0–95 years old demonstrated that vitamin D reduced the risk for acute respiratory infections overall, with the greatest benefit in those with low 25(OH)D levels (<10 ng/mL), and those receiving daily or weekly vitamin D supplementation vs. bolus doses [137,138]. Effects may be less robust in those 1–5 years old [139], and a dose effect has not been demonstrated [140]. 

Overall, these data suggest that vitamin D may have an effect in reducing the risk for respiratory infections in children, with more pronounced effects after 5 years of age, and in those that are vitamin D deficient. This effect may explain the reduced risk of asthma exacerbations in children with vitamin D supplementation (although asthma severity does not appear to improve with vitamin D). The mechanism underlying the effect on respiratory infections needs to be determined, but may involve the impact of vitamin D on inflammatory pathways.

## 3. Metabolic Syndrome and Type 2 Diabetes Mellitus

Although, VDR polymorphisms are unlikely to play a major role in obesity-related phenotypes, as reported in a population of Caucasian adults [141], vitamin D deficiency has been associated with obesity in both pediatric and adult populations. The prevalence of vitamin D deficiency is about 50% in children with obesity [142,143], and attributable to decreased sun exposure secondary to low activity level, poor nutrition with decreased consumption of vitamin D containing foods such as milk, as well as storage in adipose tissue [142]. With the increased prevalence of obesity in children there has also been concern of increased prevalence of type 2 DM (T2D), dyslipidemia and hypertension, the metabolic syndrome [1]. The role vitamin D plays in contributing to these disorders has been of interest given the above association and implications for potential treatment. Exactly which aspects of the metabolic syndrome are associated with vitamin D deficiency have not been definitively delineated as studies show varied results. Olson et al. found a negative correlation between 25(OH)D levels and HOMA-IR (homeostasis model assessment of insulin resistance) and weaker but significant inverse correlations with 2-h glucose levels in an oral glucose tolerance test (OGTT). There was no correlation with HbA1C, systolic blood pressure (SBP) or diastolic blood pressure (DBP) [142]. In another study vitamin D deficiency in children with obesity was associated with higher BMI and SBP, and decreased HDL-C [143]. After adjusting for BMI in Native American children, 25(OH)D levels were inversely associated with log transformed fasting 2-h glucose, fasting insulin, HOMA-IR, triglyceride and CRP levels, and SBP and DBP. There was a positive correlation with HDL but no correlation with total or LDL cholesterol [144]. Studies have postulated that adipokines such as adiponectin and resistin may be involved in the pathogenesis of insulin resistance and vitamin D deficiency. A study of 125 children and adolescents with obesity showed a trend toward lower 25(OH)D levels being associated with low adiponectin levels and higher insulin resistance even after adjusting for body mass, though there was no correlation between 25(OH)D levels and resistin [145]. 

Conversely, Poomthavorn et al. [146] found no correlation between vitamin D deficiency and abnormal glucose homeostasis in 150 children and adolescents with obesity living in Thailand. In fact, the five children identified with T2D in this study were all vitamin D sufficient. They also found the same degree of vitamin D deficiency in 29 healthy children without obesity [146]. Similarly, Bril et al. found no relationship between lower 25(OH)D levels and insulin resistance (using an euglycemic insulin clamp), liver fat accumulation or steatohepatitis when adult patients with the latter were matched for BMI and total adiposity with controls [147]. More recently, in a study of 215 children with T1D and 326 children with T2D vs. youth of similar age without diabetes from the 2005–2006 NHANES Survey, the prevalence of vitamin D deficiency or insufficiency did not differ in children with vs. those without diabetes [14]. Additionally, although latitude and the winter months have been associated with a higher risk of hypertension, recent studies suggest that the effect of ultraviolet light on blood pressure is likely mediated via nitric oxide synthesis and not through vitamin D production [148]. Limitations of pediatric studies include the fact that these associative studies often do not include controls matched for measures of adiposity, or use euglycemic clamps to assess insulin resistance. In addition, sample sizes are small and length of studies short. Associations of vitamin D deficiency with worsening insulin resistance or other aspects of the metabolic syndrome are yet to be proven.

Interventional studies in adults and adolescents are described in detail in Table 1 and are summarized here.

Interventional Studies in Adults with Obesity, Prediabetes and T2D: the contribution of vitamin D to the metabolic phenotype of obesity is of interest given implications for treatment. If these associations have a biological basis, repleting vitamin D should improve the phenotype. How this should be done, to what level of replacement, and the actual efficacy have been the basis of several studies. Vitamin D administration in adults has had mixed results for progression to T2D and cardio-metabolic outcomes. Among short-term studies with small numbers of participants, most have reported no changes in insulin sensitivity [35], insulin secretion [36,37], CRP [36,38], BMI [36], lipids [36,37,39], blood pressure, endothelial function, and arterial stiffness [38] following vitamin D administration, while a few have reported an improvement in insulin sensitivity [36] and blood pressure [39]. Among larger and longer-term studies, a five-year interventional study in 556 adults 25–80 years old at risk for developing T2D, reported no impact of vitamin D administration on progression to T2D, measures of glucose metabolism, serum lipids or blood pressure [40]. A subsequent pooled meta-analysis of 23 RCTs similarly found no effect of vitamin D supplementation in controlling fasting plasma glucose levels, improving insulin resistance, or preventing T2D; however, stratified analysis suggested a possible beneficial effect in those without obesity, those with prediabetes, when 25(OH)D levels were at least 20 ng/mL, and when the supplemental dose was >2000 IUs per day and given without calcium supplementation [41]. Yet, an RCT in adult participants meeting criteria for prediabetes (but not diabetes) reported no difference between groups for onset of diabetes over the study duration [42]. 

Interventional Studies in Children with Obesity, Prediabetes and T2D: similar to the adult literature, the pediatric literature includes conflicting data regarding effects of vitamin D supplementation [43,44,45,46,47,48,49,50,51,52] (Table 1). A retrospective study of vitamin D supplementation in 43 children 3–18 years old with T2D reported a decrease in BMI-SDS and HbA1C in those supplemented with vitamin D [46]. A few prospective studies in adolescents with obesity have similarly reported improved HOMA-IR and QUICKI (but not in fasting glucose, HbA1C, CRP, inflammatory markers, blood pressure) [43,45] following vitamin D administration, and an association of reductions in Apo-B and LDL-C, and increases in HDL with increases in 25(OH)D levels after supplementation [44,49]. However, most prospective interventional studies in children with obesity have failed to demonstrate a beneficial effect of vitamin D administration on fasting lipids, glucose, insulin and CRP values [47,48,50,51], and insulin secretion or sensitivity [52], and dose response studies have similarly not been able to demonstrate an effect of increasing vitamin D doses on fasting glucose, insulin and insulin resistance [48,51] or lipid levels [48]. 

Overall data indicate that there is an association between obesity and vitamin D deficiency (likely related to sequestration of vitamin D in adipose depots). However, the biological effect of vitamin D deficiency on insulin resistance, hypertension, hyperlipidemia and progression to T2D is likely small. Data are conflicting regarding associations of vitamin D deficiency with components of the metabolic syndrome, as well as a treatment effect. Some differences may be attributable to the type of study, sample size, and dose of vitamin D used (Table 1). Larger and longer-term prospective studies are needed to definitively determine causality. 

## 4. Cancer

In cell culture studies, 25(OH)D concentrations of >30 ng/mL prevent unregulated cell growth [149,150,151]. There are very little data for the role of vitamin D in preventing or ameliorating the course of cancer in children, although low 25(OH)vitamin D levels may contribute to poor bone health in children with hematological and other malignancies [152,153,154]. Much of the data for the impact of vitamin D in preventing or reducing the morbidity of cancer comes from studies in adults, particularly in the context of breast, colorectal and prostate cancer. This section briefly reviews existing literature for these three kinds of cancer. 

In breast cancer cell lines, vitamin D signaling inhibits expression of a tumor progression gene (Id1), and ablation of VDR expression causes increased tumor growth and development of metastases [155]. This pathway is inhibited in murine models of breast cancer associated with vitamin D deficiency, and epidemiological studies in humans suggest associations of vitamin D deficiency and the risk of breast, prostate and colon cancer [156,157,158]. However, data are conflicting regarding the impact of vitamin D receptor genotype and cancer risk at this time [159]. One study reported that *VDR* single nucleotide polymorphisms and haplotypes may determine how inflammatory markers change in breast cancer survivors with vitamin D deficiency, following vitamin D supplementation [160]. 

In general, vitamin D deficiency is associated with increased risk of progression and mortality in breast cancers [161,162,163]. Of note, these data have been questioned in that meta-analyses indicate that while case control studies suggest a reduction in breast cancer risk in those with higher 25(OH)D levels, this is not evident in prospective studies [163,164]. Data suggest that weight and alcohol intake may modify associations between vitamin D intake and breast cancer risk [165]. Of note, the Women’s Health Initiative demonstrated no reduction in risk for breast cancer in women receiving 400 IU vitamin D_3_ and 1000 mg of calcium vs. placebo [166]. However, a decreased risk was noted in postmenopausal women on hormone replacement therapy from daily vitamin D and calcium supplements [167], and another study reported improved breast cancer survival in patients after surgery of invasive breast cancer in de novo vitamin D users [168]. One recent 12-month RCT of 20,000 IUs of vitamin D vs. placebo in 208 premenopausal women at high risk for breast cancer found no effect of vitamin D supplementation in reducing breast cancer risk (assessed using mammographic density) [169]. Similarly, an RCT of 2000 IUs of vitamin D and omega-3 fatty acids (1 g daily) (2 × 2 factorial design) in women 55 and older and men 50 and older found no effect of vitamin D supplementation in lowering the incidence of invasive cancer compared to placebo [170].

Meta-analyses have reported inverse associations of 25(OH)D levels and vitamin D intake with the incidence and recurrence of colorectal adenoma [171,172,173,174]. One meta-analysis suggested a 10%–20% reduction in risk of incidence or recurrence of colorectal adenomas with every 20 ng/mL increase in 25(OH)D levels [171]. Another meta-analysis suggested a 26% reduction in risk with every 10 ng/mL increase in 25(OH)D levels [175,176]. However, a large Mendelian randomization study that included 10,725 colorectal cancer cases and 30,794 controls found no evidence for a causal relationship between circulating 25(OH)D and colorectal cancer risk. These authors suggested that circulating vitamin D may be a biomarker of colorectal cancer, rather than a causative factor [177]. Thus far, data for a role of VDR polymorphisms in mediating the risk of colorectal adenomas are conflicting, with no associations reported in more recent studies [176,178]. 

Some studies have suggested that vitamin D with or without calcium supplementation promotes colorectal epithelial cell differentiation, reduces proliferation, and promotes apoptosis, and is thus chemopreventive against colorectal neoplasms [179]. Further, a higher expression of CYP24A1, which reduces local 1,25-D (3) availability and thus its antiproliferative effect, has been demonstrated in colorectal adenocarcinomas, associated with increased expression of the proliferation marker Ki-67 [180]. Despite these data and consistent with the Mendelian randomization study, randomized controlled trials and meta-analyses do not support a role of vitamin D supplementation in preventing these cancers [161,181,182] or relapses [183,184,185]. Baron et al. disappointingly reported that daily supplementation with vitamin D_3_ (1000 IU), calcium (1200 mg), or both after removal of colorectal adenomas did not reduce the risk of recurrence over a 3–5 year period [184]. Similarly, Urashima et al. reported no improvement in relapse free survival in patients with digestive tract cancers (48% were colorectal cancers) randomized to 2000 IUs of vitamin D vs. placebo (AMATERASU RCT) [185]. However, in the SUNSHINE trial (a phase 2 RCT in which 139 patients with advanced or metastatic colorectal cancer were randomized to chemotherapy plus high-dose vitamin D_3_ supplementation (8000 IUs daily for one month followed by 4000 IUs daily vs. chemotherapy plus standard-dose vitamin D_3_ (400 IUs daily)) the higher vitamin D dose was associated with a multivariable hazard ratio of 0.64 for progression-free survival or death that was statistically significant [186].

Sun exposure may delay the onset of prostate cancer [157] while serum 25(OH)D levels of at least 20 ng/mL appear to reduce the risk for prostate cancer by 50% [187]. However, data for associations of vitamin D deficiency with incidence of prostate cancer are weaker than those for colon cancer, and data for associations with progression are inconsistent [161,188]. 

Overall, at this time, data do not support a significant role of vitamin D supplementation in preventing or changing the course of breast, colorectal or prostate cancer. 

## 5. Conclusions

In conclusion, vitamin D may have biological effects well beyond the skeleton. While it is important to maintain adequate 25(OH)D levels for optimal bone health, this may have benefits in a variety of different organ systems. However, interventional studies to prevent or ameliorate disease processes attributable to vitamin D deficiency in large populations have been disappointing thus far for the most part. More investigation is needed to determine optimal dosing and serum levels to effect positive biological outcomes. 

## Figures and Tables

**Table 1 nutrients-11-01460-t001:** Summary of studies of vitamin D administration in type 1 and type 2 diabetes.

Reference	Type of Study	Intervention	Participants	Results (Intervention)
**Type 1 Diabetes**				
**Studies examining effects on immune modulation**				
[25]	18-month randomized controlled trial (RCT)	2000 IUs vitamin D_3_ daily or placebo	38 participants; 35 completers; 7–30 years old	Increase in regulatory T-cell (T-reg) percentage; lower cumulative incidence of progression to undetectable C-peptide; no difference in HbA1C, insulin requirement or BMI
[26]	12-month RCT	70 IUs/kg/day vitamin D_3_ vs. placebo	29 Participants; >6 years old; <3 months duration of T1D	Increase in suppressive capacity of T-regs
**Studies of prevention of islet autoimmunity or T1D**				
[27]	Meta-analysis of four case-control and one cohort study	Vitamin D supplementation (variable doses)	Infants	Four case control studies: risk of T1D decreased; similar findings in cohort study; some evidence of a dose-response effect
[28]	Population impact number of eliminating a risk factor (PIN-ER-*t*) Statistical method	Vitamin D supplementation (variable doses)	Babies born in 2012	For a population of 729,674 babies born in England and Wales in 2012, 374 cases of T1D (out of 1357 total predicted cases) could be prevented over 18 years if all were supplemented with vitamin D
[31]	Cohort study; assessment every 3 months between 3–48 months, and then every 6 months	Maternal vitamin D supplementation during pregnancy (based on recall); cumulative intake of vitamin D supplements and n-3 FAs analyzed	8676 children with increased genetic risk for T1D in Finland, Germany, Sweden and the US	Vitamin D supplementation during pregnancy was not associated with risk for development of islet autoantibodies (any/none and cumulative intake)
**Studies examining course or complications of T1D**				
[29]	6-month RCT	0.25 mcg twice daily of alfacalcidol vs. placebo	61 participants; 54 completers; 8–15 years old; <8 weeks duration of T1D	Higher fasting C-peptide; lower daily insulin dose
[30]	12–24 weeks single arm intervention study	1000–2000 IUs of vitamin D_3_ daily	271 adolescents with T1D with 25(OH)D <15 ng/mL	Improved endothelial function; decreased urinary inflammatory cytokines/chemokines; no change in systolic or diastolic blood pressure, lipids, HbA1C and albumin/creatinine ratio
[32]	2-year RCT	0.25 mcg daily of calcitriol or placebo	34 participants 11–35 years old with recent onset T1D and high basal C-peptide	No effect on beta cell function
[33]	Single dose single arm intervention study	Vitamin D_3_: 100,000 IUs for those 2–10 years old; 160,000 IUs for those >10 years	40 children <19 years with T1D and vitamin D deficiency (<20 ng/mL) included in ITT analysis	No difference in HbA1C levels at 3 months or at 1 year
[34]	6-month RCT	Vitamin D_3_ 60,000 IUs once a month for 6 months	52 children with T1D 1–18 years old	Higher mean C-peptide level; no difference in HbA1C or insulin requirement
**Obesity, Prediabetes or Type 2 Diabetes**				
**Adults**				
[35]	Single arm intervention study	Vitamin D_3_ two doses of 100,000 IUs at 2-week intervals	33 adults with vitamin D deficiency (25(OH)D < 20 ng/mL) and without T2D	No change in mean blood glucose or insulin, or insulin sensitivity (assessed using an OGTT)
[36]	6-month RCT	Vitamin D_3_ 4000 IUs daily or placebo	82 insulin resistant, vitamin D deficient (25(OH)D < 20 ng/mL) South Asian women in New Zealand without T2D	HOMA-IR improved when 25(OH)D level rose to >32 ng/mL; no differences in insulin secretion, CRP, BMI or lipid levels
[37]	6-month intervention study	Vitamin D_3_ 20,000 IUs or placebo given twice weekly over 6 months	104 adults with vitamin D deficiency; 94 completers	No difference in insulin secretion, insulin sensitivity (using a hyperglycemic clamp) or lipids
[38]	4-month RCT	Vitamin D_3_ 2500 IUs or placebo daily	114 post-menopausal women with 25(OH)D between 10–60 ng/mL	No improvement in blood pressure, endothelial function, arterial stiffness, inflammation and CRP
[39]	16-week RCT	Vitamin D_3_ supplementation 200 IUs or placebo daily	165 healthy women 18–35 years	No change in lipids; modest change in systolic and diastolic blood pressure
[40]	5-year RCT	Vitamin D_3_ 20,000 IUs/week or placebo; followed every 6 months	556 adults 25–80 years old with prediabetes; 503 completers	25(OH)D increased from ~24 ng/mL to 48 ng/mL with supplementation; no effect on progression to T2D, measures of glucose metabolism, serum lipids or blood pressure in the group as a whole, or in those with vitamin D deficiency
[41]	Pooled meta-analysis of 28 RCTs	Vitamin D_3_ supplementation, variable doses	Adults at risk for T2D (no T2D)	No effect on controlling fasting plasma glucose levels, improving insulin resistance, or preventing T2D; stratified analysis suggested a possible beneficial effect in those without obesity, those with prediabetes, when 25(OH)D levels were ≥20 ng/mL, and when the supplemental dose was >2000 IUs per day and given without calcium supplementation
[42]	RCT	Vitamin D_3_ 4000 IUs or placebo regardless of vitamin D status	2423 adult participants meeting criteria for prediabetes (2382 randomized)	No differences in baseline 25(OH)D; supplemented group had 25(OH)D levels about twice that in the placebo group; no difference in progression to T2D (9.4 vs. 10.7 events per 100 person-years respectively at a median follow-up of 2.5 years)
**Children**				
[43]	6-month RCT	Vitamin D_3_ 4000 IUs daily or placebo	35 adolescents with obesity 9–19 years old	Improved HOMA-IR and QUICKI (but not fasting glucose, HbA1C, CRP, IL-6 or TNF-alpha) in those who received vitamin D
[44]	1-year open label parallel arm prospective study	Vitamin D_3_ 5000 IUs weekly for 8 weeks vs. no intervention	70 indigenous Argentinean children vs. 20 non-supplemented children	Improved HDL
[45]	12-week RCT	Vitamin D_3_ 300,000 IUs weekly or placebo	50 children with obesity 10–16 years old	Improved serum insulin and HOMA-IR with no effect on lipids, fasting blood sugar or blood pressure
[46]	Retrospective study	Vitamin D supplementation	43 children 3–18 years old with T2D for >12 months and a diagnosis of vitamin D deficiency (25(OH)D < 20 ng/mL)	Decrease in BMI-SDS, HbA1C and ALT in supplemented group
[47]	12-week RCT	Vitamin D_3_ 2000 IUs daily or placebo	58 adolescents with obesity 12–18 years old	No change in fasting glucose, insulin, HOMA-IR, lipids or CRP
[48]	12-week RCT	Vitamin D_3_ two doses (400 IU/day and 2000 IU/day) for 12 weeks	51 Caucasian adolescents with obesity 12–18 years old; 46 completers	No change in 25(OH)D levels in the in 400 IU/day group and a modest increase in the 2000 IU/day group. No change in fasting HOMA-IR, insulin, glucose or lipid levels post-supplementation
[49]	2-year prospective study	Vitamin D_3_ 100,00 IUs/year to both groups	104 children in Group A (treated in 2014) and 86 in Group B (treated in 2013)	Changes in 25(OH)D levels were significantly associated with lower LDL-C and Apo-B levels.
[50]	3-month open label, prospective study	Vitamin D_3_ 100,000 IUs monthly for 3 months	19 children with obesity and vitamin D deficiency 13–18 years old	No change in endothelial function, fasting lipids, glucose, insulin and CRP values
[51]	12-week RCT	Vitamin D_3_ at either 0, 400, 1000, 2000 or 4000 IU/day for 12 weeks	323 early pubertal children At baseline, 15% had 25(OH)D levels that were insufficient <25 ng/mL, 6% had levels <16 ng/mL and 1% had levels lower than 12 ng/mL	At baseline, 25(OH)D levels were inversely associated with insulin and HOMA-IR. However, glucose, insulin and insulin resistance increased over 12 weeks in all dosage groups
[52]	12-week RCT	Vitamin D_3_ 50,000 IUs per week vs. placebo	29 African American children with obesity 13–17 years old	No impact on insulin secretion or sensitivity

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
