# Peer review of "Extra-Skeletal Effects of Vitamin D"

_nutrients, 2019, doi:10.3390/nu11071460_

Round 1

Reviewer 1 Report

It was not obvious what the aims of this review were.  Although it covered a wide area, it was at a rather superficial level and was certainly not comprehensive with many important studies and reviews omitted.  The reasons for including details of some articles but not others were not explained.  In addition the review was not up-to-date and, in the area of vitamin D research which is developing so quickly, this was not acceptable.  Indeed out of 109 references, only one was published in 2019, one in 2018, none in 2017 and six in 2016.  Furthermore no criticisms were made of the described studies and reasons suggested for what were frequently contradictory results.  Perhaps the authors could consider concentrating on studies that include children only which would make the review more focussed and useful, especially for paediatricians.  In addition it would make it different from the many other reviews on the topic of vitamin D and its extra-skeletal effects. 

Smaller points follow:

 The Abstract gives no results or conclusions.

The Introduction is an exact repeat of the Abstract.  At least information is required about the metabolic pathways leading from UV irradiation of the skin to the active form of vitamin D and how the vitamin D status of an individual is assessed.  

Line 28.  References 1 and 2 are totally outdated.

Line 36-37.  A reference is required here.

Line 46.  The "groups" are not defined.

Lines 44-66 and elsewhere.  In place of using text to describe studies in a particular area, constructing a Table giving all the relevant details is recommended.

Lines 95-97.  References 34, 35 and 36 are muddled up.

LIne 106.  This section fails to mention that the symptoms of SLE can worsen after UV radiation exposure which will increase the vitamin D status of the individual.  See Shoenfeld et al 2018 for a recent review.

LIne 153,  This section fails to mention that there is no correlation between the increase in 25(OH)D post phototherapy and less severe symptoms of psoriasis.

LIne 157.  No mention is made in this section of the recent evidence that the reduction in hypertension following UVR exposure is not through vitamin D but probably through the release of NO in the skin.  In addition there is no strong association between obesity and genetic variants in the VDR - see Correo-Rodriguez et al 2018.

Line 254.  This section is particularly out-of-date.  As an example, in PubMed, more than 200 articles on the topic of vitamin D/cancer have been published in the first five months of 2019.

Author Response

We thank the Reviewer for the thorough review of the manuscript and have made necessary changes to the content based on the comments. We believe the manuscript has been strengthened significantly in the process, and hope that it is now suitable for publication.

General Comments: It was not obvious what the aims of this review were.  Although it covered a wide area, it was at a rather superficial level and was certainly not comprehensive with many important studies and reviews omitted.  The reasons for including details of some articles but not others were not explained.  In addition the review was not up-to-date and, in the area of vitamin D research which is developing so quickly, this was not acceptable.  Indeed out of 109 references, only one was published in 2019, one in 2018, none in 2017 and six in 2016.  Furthermore no criticisms were made of the described studies and reasons suggested for what were frequently contradictory results.  Perhaps the authors could consider concentrating on studies that include children only which would make the review more focussed and useful, especially for paediatricians.  In addition it would make it different from the many other reviews on the topic of vitamin D and its extra-skeletal effects. 

Response to General Comments: We have now included the intent of the manuscript in the Introduction section. A significant amount of material has now been added to the various sections to ensure that these sections are now up to date. We have also added information regarding possible reasons for conflicting results. We have included studies from children wherever available, but also describe key studies in adults when data are lacking in children, or for conditions more common in adults than children.

Comment 1: The Abstract gives no results or conclusions.

Response to Comment 1: Given that this is a review with many studies, it is not possible to summarize the results in their entirety- we have summarized the findings in the Abstract.

Comment 2: The Introduction is an exact repeat of the Abstract.  At least information is required about the metabolic pathways leading from UV irradiation of the skin to the active form of vitamin D and how the vitamin D status of an individual is assessed.  

Response to Comment 2: The Introduction has been modified as recommended.

Comment 3: Line 28.  References 1 and 2 are totally outdated.

Response to Comment 3: We have now added more recent reference to this section.

Comment 4: Line 36-37.  A reference is required here.

Response to Comment 4: We have added a reference as recommended.

Comment 5: Line 46.  The "groups" are not defined.

Response to Comment 5: We have now clarified this.

Comment 6: Lines 44-66 and elsewhere.  In place of using text to describe studies in a particular area, constructing a Table giving all the relevant details is recommended.

Response to Comment 6: We have now included a Table (Table 1) detailing studies pertinent to obesity, prediabetes, Type 2 diabetes and Type 1 diabetes.

Comment 7: Lines 95-97.  References 34, 35 and 36 are muddled up.

Response to Comment 7: We have now made sure that all references are correct.

Comment 8: LIne 106.  This section fails to mention that the symptoms of SLE can worsen after UV radiation exposure which will increase the vitamin D status of the individual.  See Shoenfeld et al 2018 for a recent review.

Response to Comment 8: We thank the Reviewer for the suggestion and have added this.

Comment 9: LIne 153,  This section fails to mention that there is no correlation between the increase in 25(OH)D post phototherapy and less severe symptoms of psoriasis.

Response to Comment 9: We have now included this reference as recommended.

Comment 10: LIne 157.  No mention is made in this section of the recent evidence that the reduction in hypertension following UVR exposure is not through vitamin D but probably through the release of NO in the skin.  In addition there is no strong association between obesity and genetic variants in the VDR - see Correo-Rodriguez et al 2018.

Response to Comment 10: We have now included this reference as recommended.

Comment 11: Line 254.  This section is particularly out-of-date.  As an example, in PubMed, more than 200 articles on the topic of vitamin D/cancer have been published in the first five months of 2019.

Response to Comment 11: We have now updated this section extensively

Reviewer 2 Report

Dear Editor,

the manuscript “Extraskeletal effects of vitamin D” is a well-written review exploring actual literature regarding the expanding role of vitamin D beyond bone health.

The review is divided in two main chapters: “Immune and anti-inflammatory effects” and “Metabolic syndrome and Type 2 diabetes mellitus”. The first chapter describes the role of vitamin D in Type 1 diabetes, multiple sclerosis, rheumatoid arthritis, SLE and JDM, IBDs, food allergies, chronic hepatitis, and psoriasis. I think that the authors should resume at the end of every section their opinion regarding if vitamin D plays or not a significant role for every condition described (as they have done at the end of the section on Type 1 diabetes). Moreover, the authors did not review the role of vitamin D in asthma and respiratory tract infections. Indeed, in these conditions recent meta-analyses suggested an important role of vitamin D, particularly in pediatric age and in subjects with severe vitamin D deficiency.

Regarding the second part “Metabolic syndrome and Type 2 diabetes mellitus”, the authors may prepare a table to resume the results of the cited studies of vitamin D supplementation and make reading more comfortable.

Finally, abstract is practically identical to introduction, so I suggest some changes to differentiate the two paragraphs.

Author Response

We thank the Reviewer for the thorough review of the manuscript and have made necessary changes to the content based on the comments. We believe the manuscript has been strengthened significantly in the process, and hope that it is now suitable for publication.

Comment 1: The review is divided in two main chapters: “Immune and anti-inflammatory effects” and “Metabolic syndrome and Type 2 diabetes mellitus”. The first chapter describes the role of vitamin D in Type 1 diabetes, multiple sclerosis, rheumatoid arthritis, SLE and JDM, IBDs, food allergies, chronic hepatitis, and psoriasis. I think that the authors should resume at the end of every section their opinion regarding if vitamin D plays or not a significant role for every condition described (as they have done at the end of the section on Type 1 diabetes). Moreover, the authors did not review the role of vitamin D in asthma and respiratory tract infections. Indeed, in these conditions recent meta-analyses suggested an important role of vitamin D, particularly in pediatric age and in subjects with severe vitamin D deficiency.

Response to Comment 1: Based on the Reviewer’s comments, we now end each section with a summary of studies to date and our overall impression of whether or not vitamin D supplementation is beneficial in that condition

Comment 2: Regarding the second part “Metabolic syndrome and Type 2 diabetes mellitus”, the authors may prepare a table to resume the results of the cited studies of vitamin D supplementation and make reading more comfortable.

Response to Comment 2: We have now done this for all interventional studies in participants with type 1, and those with obesity, prediabetes and type 2 diabetes (Table 1)

Comment 3: Finally, abstract is practically identical to introduction, so I suggest some changes to differentiate the two paragraphs.

Response to Comment 3: We have now modified the abstract and the introduction and they are no longer identical

Round 2

Reviewer 1 Report

The manuscript is significantly improved by the addition of data from more recent references throughout, an explanation of the aims of the review, the construction of the Table and a summary of the contradictory results at the end of each section.   The review now gives a nicely balanced and well written account of the current controversies surrounding the potential role of vitamin D in extra skeletal diseases.  Two minor comments remain:

 Line 35.  Suggest that the enzymes involved in the synthesis and degradation of 1,25(25)2D are mentioned as they are referred to below in lines 58-71 and elsewhere in the review.  

Table 1.  Suggest that the text which gives details of the studies outlined in Table 1 could be shortened as it is largely a repeat of what is in the Table.  The text should then give only the overall conclusions based on the data in the Table.  The references in Table 1 should be as numbers not as authors' names and date of publication.     

Author Response

Overall Comments: The manuscript is significantly improved by the addition of data from more recent references throughout, an explanation of the aims of the review, the construction of the Table and a summary of the contradictory results at the end of each section.   The review now gives a nicely balanced and well written account of the current controversies surrounding the potential role of vitamin D in extra skeletal diseases.  

Response to Comment 1: We thank the Reviewer for the positive comments, and appreciate the suggestions to improve the manuscript

 Minor Comment 1: Line 35.  Suggest that the enzymes involved in the synthesis and degradation of 1,25(25)2D are mentioned as they are referred to below in lines 58-71 and elsewhere in the review.  

Response to Minor Comment 1: We have now added this information as recommended.

Minor Comment 2: Table 1.  Suggest that the text which gives details of the studies outlined in Table 1 could be shortened as it is largely a repeat of what is in the Table.  The text should then give only the overall conclusions based on the data in the Table.  The references in Table 1 should be as numbers not as authors' names and date of publication.     

Response to Comment 2: Based on the recommendations of the Reviewer, we have shortened the associated text significantly, while maintaining the context of the material.